# Comparative Analysis of Posiphen Pharmacokinetics across Different Species—Similar Absorption and Metabolism in Mouse, Rat, Dog and Human

**DOI:** 10.3390/biom14050582

**Published:** 2024-05-15

**Authors:** Maria L. Maccecchini, Diane R. Mould

**Affiliations:** 1Annovis Bio, Inc., 101 Lindenwood, Malvern, PA 193551, USA; 2Projections Research Inc., 535 Springview Lane, Phoenixville, PA 19460, USA; drmould@pri-home.net

**Keywords:** posiphen, pharmacokinetics, metabolism, neurodegenerative diseases, neurodegeneration, species comparison

## Abstract

Posiphen is a small molecule that exhibits neuroprotective properties by targeting multiple neurotoxic proteins involved in axonal transport, synaptic transmission, neuroinflammation, and cell death. Its broad-spectrum effects make it a promising candidate for treating neurodegenerative conditions, including Alzheimer’s and Parkinson’s diseases. Despite extensive investigation with animal models and human subjects, a comprehensive comparative analysis of Posiphen’s pharmacokinetics across studies remains elusive. Here, we address this gap by examining the metabolic profiles of Posiphen and its breakdown into two primary metabolites—N1 and N8—across species by measuring their concentrations in plasma, brain, and CSF using the LC-MS/MS method. While all three compounds effectively inhibit neurotoxic proteins, the N1 metabolite is associated with adverse effects. Our findings reveal the species-specific behavior of Posiphen, with both Posiphen and N8 being predominant in various species, while N1 remains a minor constituent, supporting the drug’s safety. Moreover, in plasma, Posiphen consistently showed fast clearance of all metabolites within 8 h in animal models and in human subjects, whereas in CSF or brain, the compound has an extended half-life of over 12 h. Combining all our human data and analyzing them by population pharmacokinetics showed that there are no differences between healthy volunteers, Alzheimer’s, and Parkinson’s patients. It also showed that Posiphen is absorbed and metabolized in a similar fashion across all animal species and human groups tested. These observations have critical implications for understanding the drug’s safety, therapeutic effect, and clinical translation.

## 1. Introduction

Posiphen (also known as posiphen tartrate or buntanetap) is a hydrophobic small molecule with oral bioavailability and blood–brain barrier permeability; it is a translational inhibitor of neurotoxic aggregating proteins (TINAPs), including amyloid precursor protein (APP) [1,2,3,4,5,6,7,8], tau [8], α-synuclein (αSYN) [9,10], and huntingtin (HTT) [11]. By inhibiting more than one neurotoxic protein, we believe that it may show better efficacy in a number of neurodegenerative diseases. Our previous investigations, conducted in both animal models and human subjects, have demonstrated that Posiphen improves the affected functions across a diverse spectrum of conditions, including Alzheimer’s disease [7,12,13], Parkinson’s disease [10,12], stroke [14,15], traumatic brain injury [16], and Huntington’s disease [11]. Currently, Posiphen is under investigation in humans with moderate Alzheimer’s (Phase 2/3) (NCT05686044) and early Parkinson’s disease (Phase 3) (NCT05357989).

While we have assembled a large amount of data on the bioavailability and distribution of the drug, a systematic comparative analysis of Posiphen’s pharmacokinetics across different species, blood–brain barrier penetration, and metabolic degradation has not been described. Herein, we decided to conduct an extensive and rigorous comparative analysis across different species and in different human populations.

This manuscript seeks to assess the pharmacokinetics (PK) of Posiphen and its metabolites in mice, rats, dogs, and in healthy humans, as well as Alzheimer’s and Parkinson’s patients, thereby advancing our understanding of Posiphen’s translational potential in the context of neurodegenerative diseases. 

## 2. Materials and Methods

### 2.1. Investigational Drug

Posiphen, (3aR)-1, 3a, 8-trimethyl-1, 2, 3, 3a, 8, 8a-hexahydropyrrolo (2, 3-b) indol-5-yl phenylcarbamate tartrate (investigational new drug #72 654) was manufactured according to Good Manufacturing Practice (GMP) regulations (Wilmington PharmaTech, Newark, DE, USA). 

Standards of deuterated Posiphen and the two N1- and N8- norposiphen metabolites were synthesized by Chemtos (Round Rock, TX, USA) to >99% purity and used in the PK analysis conducted by Charles River Laboratories.

### 2.2. Animal Models

The animal PKs were taken from multiple GLP toxicology studies, biomarker studies, and efficacy studies [7,8,10,12]. Briefly, the data from the following species and respective Posiphen dosage levels were collected for this manuscript: CD-1 mice (*n* = 7)—65 mg/kg; Sprague Dawley rats (*n* = 6)—40 mg/kg; Beagle dogs (*n* = 12)—20 mg/kg. Sample size (*n*) represents the number of animals at each time point of sample collection. Posiphen was either given as a capsule to dogs or as gavage to mice and rats. Dose levels were chosen to characterize the pharmacokinetics of Posiphen in respective species over the desired dose/concentration range. The dose volume was determined according to animals’ weight collected on the dosing day.

### 2.3. CYP Phenotyping

In the manuscript, we describe the metabolism of Posiphen and its interaction with Cytochrome P450 enzymes which were measured in different tests across multiple studies. Briefly, CYP metabolism was tested by seven isozymes in two ways. In the first, Posiphen was incubated with human liver microsomes alone and in the presence of isozyme-specific inhibitors. The change in Posiphen’s half-life was measured—an increase in the presence of an inhibitor implied metabolic involvement by that isozyme. In the second CYP reaction phenotyping assay, Posiphen was incubated with Supersomes (recombinant CYP isozymes), where a decrease in Posiphen’s half-life in the presence of a specific recombinant CYP isozyme implied metabolic involvement by that isozyme. CYP induction was measured by taking 3 human hepatocyte sandwich-cultures that had been cryopreserved with three concentrations of Posiphen for two consecutive 24 h periods, followed by measurement of three CYP isozymes (1A2, 2B6, 3A4) via isozyme-specific probe metabolism. The gene expression of these three isozymes after Posiphen exposure was also measured. The FDA considers ≥ 2-fold change in any concentration, in any donor, in either activity or gene expression, to be evidence of induction.

### 2.4. Participants

This paper describes data from multiple studies. The healthy volunteers’ data come from the single ascending and multiple ascending dose study, whereas Alzheimer’s and Parkinson’s patients are represented by the MCI NCT01072812 study, ADCS NCT02925650 study, and AD/PD NCT04524351 study [8,12,13]. The evaluations are based on a sample size of *n* = 4 for each time point. Posiphen was administered in an immediate release solid oral dosage form, prepared in hard capsule shells, manufactured in accordance with GMP regulations by Frontage Laboratories (Exton, PA, USA). 

### 2.5. Sample Collection

In mice, blood samples were collected via the saphenous vein or other suitable blood collection site into a microcentrifuge tube containing the anti-coagulant K_2_-EDTA on wet ice and processed for plasma. For samples collected within the first hour of dosing, ±1 min is acceptable, while for all other timepoints, samples were taken within 5% of the scheduled time and are not considered as protocol deviations. Plasma samples were centrifuged (3200× *g* for 10 min at 4 °C) within an hour of collection. CSF samples were collected with 3 μL volume per time point, frozen on dry ice, and stored at −60 °C for further analysis. For brain collection in mice, animals were euthanized using CO_2_ inhalation. Whole brain was collected, washed in cold saline, dried, and homogenized immediately at 70 ± 10 °C. Homogenizing buffer MeOH/15 mM PBS (1:2, *v*:*v*) was used at a ratio of 1:9 (1 g of tissue with 9 mL of buffer; the dilution ratio is 10).

In rats and dogs, blood samples were centrifuged, and the resultant plasma was separated, and 200 µL (rat) or 400 µL (dog) of plasma was transferred to uniquely labeled pre-chilled polypropylene tubes containing 5 µL (rat) or 10 µL (dog) Dichlorvos spiking solution (nominal concentration of 0.82 mg/mL in MeOH/Water, 20/80, *v*/*v*). Plasma samples were analyzed for concentrations of Posiphen using a validated LC-MS/MS method and for concentrations of N1-norposiphen and N8-norposiphen using qualified LC-MS/MS with a LLOQ of 0.5 ng/mL for all 3 analytes.

In humans, plasma and lumbar CSF samples were collected via an indwelling catheter over 12 h (at 0, 1, 1.5, 2, 3, 4, 6, 8, and 12 h) initiated at the same time of day, 1 day prior to the start of dosing, to obtain time-dependent baseline control data, and then at the exact same times immediately after the last dose was administered. Samples were frozen and then stored at −80 °C. CSF and plasma samples were matched and analyzed for pharmacokinetics of Posiphen and metabolites (N1-norposiphen, N8-norposiphen, and N1,N8-norposiphen).

### 2.6. Posiphen Pharmacokinetics 

Concentrations of Posiphen, N1-norposiphen, N8-norposiphen, and N1, N8-norposiphen in human plasma and CSF, as well as animals’ plasma, brain, and CSF samples, were measured by high-performance liquid chromatography with the tandem mass spectrometry (LC-MS/MS) detection method, validated at Charles River Laboratories (Montreal ULC). In humans, the bioanalytical method was performed in accordance with current FDA, Industry Guidelines, and OECD Principles. The determination of Posiphen in human plasma was performed using an assay range of 0.100 to 150 ng/mL. The sample analysis was conducted in accordance with current GCP and GLP principles, and the results were presented for pharmacokinetic profiling (Absorption Systems, Exton, PA, USA). Calibration ranges for each analyte ranged from 1000 ng/mL to 1 ng/mL (or ng/g for brain) in plasma, brain, and CSF matrices. The detection limit was 0.025 ng/mL. The following PK parameters were collected for the purpose of this manuscript: maximum concentration (Cmax), time to maximum drug concentration (Tmax), half-life (T1/2), and the Area Under the Curve (AUC). 

### 2.7. Statistical Analysis 

All assay data collected were analyzed using a repeated measures mixed model analysis of variance and presented as mean values of each time-point group. Descriptive statistics—sample size (*n*), arithmetic mean, standard deviation (SD), and coefficient of variation expressed as a percent (%CV)—were generated. Compound symmetry was assumed as an appropriate covariance pattern between observations on the same patient, which provided a reasonable model fit. Assumptions of constant variance, normality or residuals and parallelism were used to assess the acceptability of the statistical model. Data are presented as means with SD. The statistical evaluations were undertaken by Data Magik (Salisbury, UK). Due to the low number of human subjects (*n* = 4), the study was not sufficiently powered to perform statistical analysis and, hence, only mean values for Cmax are presented. 

### 2.8. Population PK Analysis

Population pharmacokinetic analysis was conducted using data from healthy volunteers from Study 101 (single ascending dose study with 72 subjects) and Study 102 (multiple dose study with 48 volunteers) and in patients (multiple dose study in 68 patients with early Alzheimer’s or early Parkinson’s disease) [8,12]. Data were fit using a Bayesian approach as implemented in the software package Nonmem^®^ (version 7.5.1, Icon Development Solutions, Dublin, Ireland). Standard model building approaches were used [17]. After a base model was established, covariates were evaluated singly and assessed graphically using eta plots—The full model consisted of all statistically significant covariates identified during the single covariate evaluation phase, and back elimination was conducted. Each covariate was removed from the full model, and the impact of that removal was assessed based on model output. The final model performance was assessed graphically and using simulation with a visual predictive check (VPC) [18]. A detailed methodology is described in Appendix A.

## 3. Results

### 3.1. PK of Posiphen across Species

We initiated our study by assessing the plasma pharmacokinetics (PK) of Posiphen across various species—mice, rats, dogs, and humans. Posiphen was administered orally (PO) at the following concentrations—mice (65 mg/kg), rats (40 mg/kg), dogs (20 mg/kg), and humans (80 mg/day (e.g., approximately 1.1 mg/kg)). Due to minimal difference in Posiphen uptake between female and male animals for each species, the current analysis included mixed-gender groups. Blood plasma samples were collected at various time points within the initial 8 h following administration (6 h for humans). Plasma concentrations (ng/mL) were measured over the 8 h observation window to characterize the maximum concentration (Cmax), the time to peak drug concentration (Tmax), the half-life (T1/2), and the Area Under the Curve (AUC). The results indicated distinct pharmacokinetic profiles for Posiphen across the tested animal species consistent with the animal’s size and weight.

The T1/2 was 0.562 h for mice, 1.03 h for rats, 1.96 h for dogs, and 4.04 h for humans. As expected, the half-life increased with weight (0.56 h for mice, 1.03 h for rats, 1.96 h for dogs, and 4.04 h for humans). After 8 h, all species exhibited near-complete clearance of Posiphen from their systems. The Tmax also increased with size (0.5 for mice; 0.5 for rats; 1.18 for dogs; 1.45 for humans), and the Cmax and AUC depended on the administered dose (Table 1; Figure 1).

### 3.2. PK of Posiphen and Metabolites in Mice and Humans

As our next step, we assessed the distribution of Posiphen and its metabolites in mice and in humans in plasma, brain, and CSF (Table 2). Posiphen undergoes first-pass metabolism into N1-norposiphen (N1) and N8-norposiphen (N8) by liver cytochrome P450 (CYP) isozyme CYP3A4 (Figure 2).

Previously, we evaluated CYP metabolism across seven isozymes, with Posiphen incubated with human liver microsomes both alone and in the presence of isozyme-specific inhibitors [*pers. comm*.]. An increase in Posiphen’s half-life in the presence of an inhibitor suggests the involvement of that particular isozyme in its metabolism. In human liver microsomes, Posiphen exhibited a half-life of 27 min influenced only by CYP3A4, as evidenced by a half-life of 238 min in the presence of ketoconazole, a CYP3A4 inhibitor. Control experiments validated the assay’s success, indicating that Posiphen undergoes first-pass metabolism by liver CYP3A4 and that this metabolism is inhibited by ketoconazole.

Posiphen and the N8 metabolite have TINAP and no acetylcholinesterase inhibitory (AChEI) activity, while the N1 metabolite has both TINAP and AChEI activities, with the latter being linked with such side effects as nausea and vomiting, at the maximum tolerated dose of 160 mg [8]. Therefore, the evaluation of metabolites was the next logical step in our understanding of Posiphen’s PK and its effects across various models, both preclinical and human. The N1,N8-norposiphen metabolite was minor and hence was only measured in humans.

#### 3.2.1. Plasma

In mice, Posiphen reached its Cmax of 939 ng/mL at a Tmax 0.5 h and then rapidly metabolized into N1 (Cmax 554 ng/mL) with Tmax at 1.25 h and to N8 (Cmax 1931 ng/mL) with a Tmax at 1.5 h. The primary metabolite detected in plasma was N8, which achieved a level 1.9 times higher than Posiphen and 3.8 times higher than N1, while the N1 metabolite only reached half the level of Posiphen and a fourth of the level of N8 (Figure 3a). In humans, Posiphen is also metabolized by the liver CYP3A4, but to a lesser extent than in mice [19,20]. In our human samples, Posiphen emerged as the predominant species in plasma (Cmax 117 ng/mL) at 1.5 h, succeeded by N8 (Cmax 30.4 ng/mL) at 2 h and subsequently by N1 (Cmax 24.9 ng/mL) also at 2 h (Figure 3b). The relative level of N8 and N1 is 32% and 24% of Posiphen, respectively. This observation is highly favorable, as the lower concentration of N1 is linked to a more favorable safety profile for Posiphen, attributable to the AChEI activity exhibited by the N1 metabolite.

#### 3.2.2. CSF/Brain

Further, we sought to extend our understanding of Posiphen PK by evaluating the blood–brain barrier (BBB) permeability and the levels of Posiphen in brain and CSF in comparison with plasma. The data are time-matched and recorded at similar intervals for 12 h post-dose. This approach allows us to explore the pharmacokinetic profiles in a more clinically relevant context, providing critical insights that bridge the gap between preclinical models and human subjects.

In mice, consistently across all sample types, N8 exhibited the highest concentration, followed by Posiphen and then N1 (Figure 4a,b). Within the CSF, N8 reached its Cmax 309 ng/mL at 2 h, while Posiphen achieved 163 ng/mL at 0.5 h, and N1 reached 54.4 ng/mL at 0.5 h. In brain samples, mirroring this trend, the Cmax for N8 was 7293 ng/mL at 2 h, surpassing Posiphen at 4737 ng/mL with a Tmax of 0.5 h and N1 at 1854 ng/mL with a Tmax of 1 h. This consistent pattern across diverse sample types underscores the species-specific pharmacokinetic profiles of N8, Posiphen, and N1 in mice. The concentration of Posiphen and its metabolites in CSF in human samples revealed a somewhat similar pharmacokinetic profile with the mouse data (Figure 4c). The predominant species was N8, reaching its Cmax of 5.08 ng/mL at 3 h, followed by N1 with 2.56 ng/mL at Tmax 4 h and Posiphen with 2.23 ng/mL at Tmax 3 h.

Subsequently, we calculated the maximum concentration ratios for CSF and brain compared to plasma values for each compound to gain deeper insights into the drug’s permeability within the nervous system. Posiphen is very hydrophobic (LogD = 2.2) and easily partitions into fatty tissues like brain, leading to high brain levels. Subsequently, it leads to low CSF levels, since CSF is very aqueous and contains no fat.

Overall, in mice, the CSF/plasma values for all compounds were low (<1). Posiphen exhibited a ratio of 0.17 [163 ng/mL:939 ng/mL], the highest value when compared to the other two metabolites. For N8, the CSF-to-plasma ratio was 0.16 [309 ng/mL:1931 ng/mL], while for N1, it was 0.10 [54.5 ng/mL:554 ng/mL]. In contrast, the ratios of concentrations in brain vs. plasma exhibited markedly higher values for all compounds, supporting our earlier data for blood–brain barrier permeability. Specifically, Posiphen demonstrated a ratio of 5.04 [4737 ng/mL:939 ng/mL], followed by N8 with a ratio of 3.78 [7293 ng/mL:1931 ng/mL] and then by N1, which had the lowest brain-to-plasma concentration proportion value at 3.35 [1854 ng/mL:554 ng/mL]. These findings underscore the differential distribution of the compounds between the central nervous system and plasma. The human samples revealed the highest ratio value for N8, which was 0.17 [5.08 ng/mL:30.4 ng/mL]. It was followed by the N1 metabolite at a ratio of 0.10 [2.56 ng/mL:24.9 ng/mL] and finally by Posiphen with a ratio of 0.02 [2.23 ng/mL:117 ng/mL], which had the lowest CSF-to-plasma values among all three compounds. These calculated ratios substantiate and align closely with the experimental data obtained from mice, reinforcing the observation of consistently low concentrations of these compounds within the cerebrospinal fluid.

### 3.3. Population PK (popPK) in Human Healthy Volunteers and Alzheimer’s and Parkinson’s Patients

As the next logical step following Posiphen PK across animal species and humans, we looked at possible variability among several clinical subgroups. Population pharmacokinetics included multiple healthy volunteers and patients with early Alzheimer’s or early Parkinson’s diseases. The analysis revealed a direct correlation of the administered dose of Posiphen with the Michaelis constant, relative bioavailability, and absorption rate constant (Figure 5). As the dose increased from 0.04 to 2.4 mg/kg, the Michaelis constant (km) increased from 10 to 15, and the absorption rate constant (Ka) increased from 3.5 to 6 1/h. For the relative bioavailability, as the dose increased from 0.04 to 2.4 mg/kg, the bioavailability increased from 0.4 to 1.7. For the central volume, as the body mass index increased from 20 to 45 kg/m^2^, the central volume decreased from 900 to 200 L. The presented data comprise a collection of Posiphen PK in different subgroups and show no variability in the drug’s absorption, metabolism, and clearance between healthy subjects and patients with either Alzheimer’s or Parkinson’s diseases.

## 4. Discussion

The data presented in this study revealed the overall species-specific behavior of Posiphen while demonstrating certain similarities in its absorption and metabolism in various models. In all analyzed species—mouse, rat, dog, and human—Posiphen reaches its peak within the first 2 h. It is important to note that in smaller species (mouse and rat), this peak was reached in under 1 h, and in larger species (dog and human), between 1 and 2 h. Moreover, despite weight-dependent variations in half-life, Posiphen showed a rapid clearance from plasma (within 8 h) in all studied animal models, underscoring the uniformity of the drug’s metabolism. This effect is likely a key factor in the drug’s safety profile, as demonstrated in our completed clinical trials in Alzheimer’s and Parkinson’s patients [7,8,12]. Another point of similarity is the permeability of Posiphen into the brain and CSF, as witnessed in mice and humans. Due to the hydrophobic nature of Posiphen, it easily partitions in the fatty tissue, increasing its concentration in the brain as compared to CSF. The brain pharmacokinetics of a centrally acting drug such as Posiphen and thereby its associated pharmacodynamics are the result of a combination of influx and efflux processes plus the extent of distribution of the drug within the brain tissue. Our results reaffirm the partitioning of a hydrophobic molecule such as Posiphen into fatty tissue and not into aqueous solutions. Additionally, it supports the ability of the drug to cross the BBB, which likely accounts for its efficacy in improving cognitive functions in mice [7] and in patients with Alzheimer’s and Parkinson’s [8,12]. 

However, these observations must be approached with caution because we also observed distinct species-specific behaviors. First, the metabolic profile in plasma is different between species. Posiphen emerged as the primary compound in the plasma of humans and rats, while it was N8 in mice and N1 in dogs (Appendix A). These differences derive from the species variations in the CYP3A4 family of liver isoenzymes. In humans, Posiphen is metabolized by CYP3A4, while in mice, the equivalent of the CYP3A4 isoform is CYP3A1, which is expressed in both the liver and the small intestine. Mice have very high levels of CYP3A1, which has 76% amino acid homology with human CYP3A4, and hence, they metabolize Posiphen faster than the other species. Interestingly, these differences disappear in CSF/brain, where the dominant species is N8. Second, we observed the long persistence of Posiphen and N8 in the human CSF with a half-life of >12 h; accordingly, we saw the long half-life of N8 in mice. This observation and the exact reasoning behind this human-specific persistence is to be investigated in our future studies.

Finally, the popPK clearly shows that Posiphen is absorbed, distributed, metabolized, and secreted in a very similar fashion in healthy volunteers and Alzheimer’s and Parkinson’s patients. No variation was observed for people with liver or kidney issues or by weight. This follows the trend that we see in animal studies on the consistency of Posiphen’s metabolic profile despite the disease state. This is an important observation which reaffirms the consistency of certain Posiphen behavior in the organism, not only in preclinical and translation stages, but also in real-world clinical settings. We will continue to investigate the popPK data from our ongoing Alzheimer’s Phase 2/3 (NCT05686044) and Parkinson’s Phase 3 (NCT05357989) studies, which might provide new insights into potential variations for certain disease subgroups.

## 5. Conclusions

Our study aimed to provide a comprehensive comparison of Posiphen PK data collected over time across multiple studies involving several animal species and human subjects. The primary goal was to understand the differences in Posiphen behavior that could have significant clinical implications. Despite certain species-specific variations, our overall observation revealed a remarkable consistency, with Posiphen reaching its peak plasma concentration rapidly, followed by high brain/CSF absorption and then complete clearance. These findings contribute valuable insights to the field of drug development and reinforce the potential of Posiphen as a safe therapeutic option for such neurodegenerative diseases as Alzheimer’s and Parkinson’s.

## Figures and Tables

**Figure 1 biomolecules-14-00582-f001:**
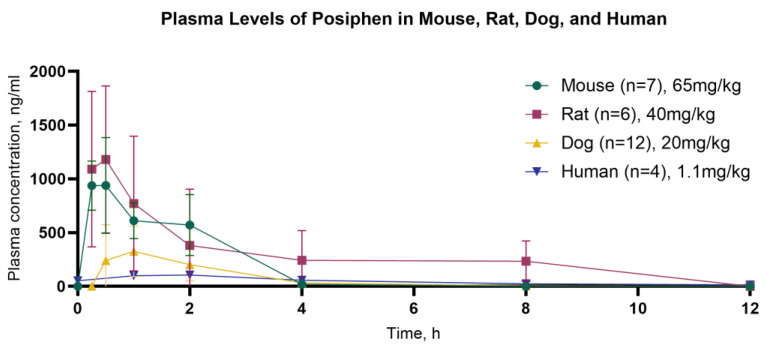
Time curve for mean plasma concentration of Posiphen over the 12 h period following oral administration in mouse, rat (n = 6), dog, and human (n = 4). The SD and %CV values for each time point are described in Appendix A.

**Figure 2 biomolecules-14-00582-f002:**
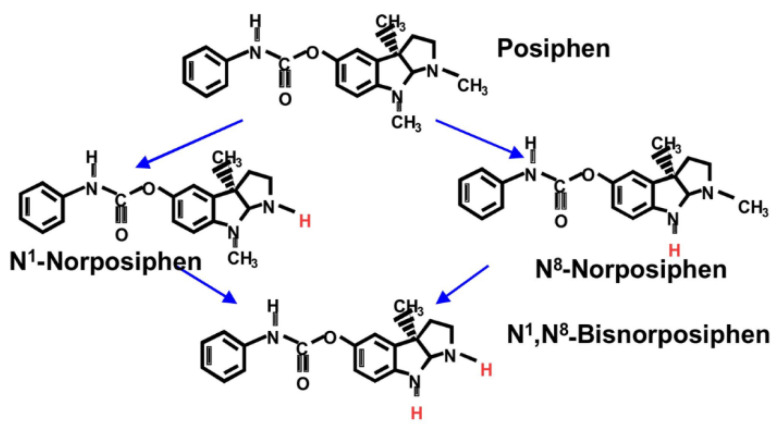
Schematic of CYP-mediated Posiphen metabolism into metabolites: N1-norposiphen, N8-norposiphen, and N1,N8-norposiphen.

**Figure 3 biomolecules-14-00582-f003:**
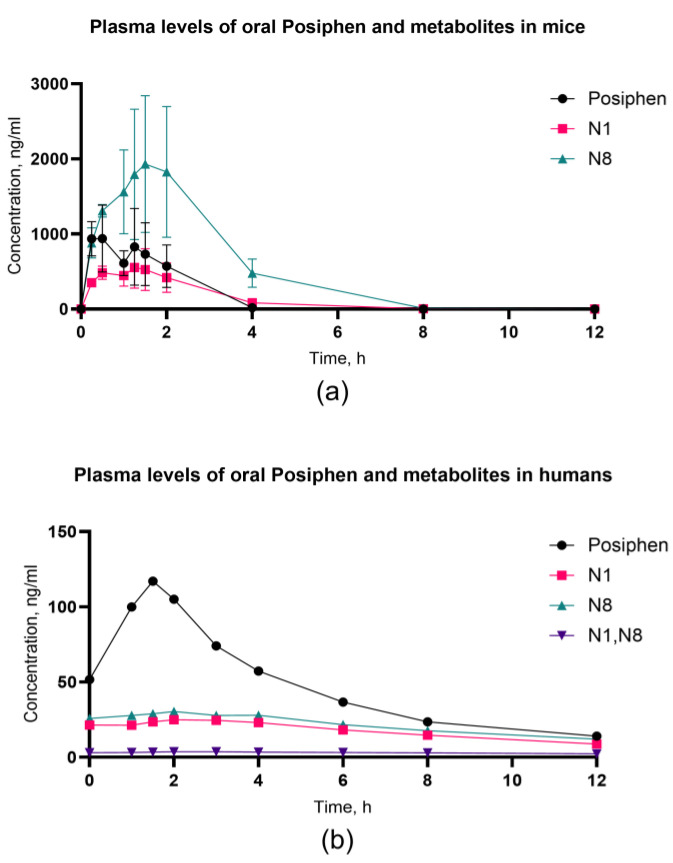
Plasma concentrations of oral Posiphen and metabolites presented as Cmax mean values with standard deviation (SD) in (**a**) mice (*n* = 7) and (**b**) humans (*n* = 4). The SD and %CV values for each time point in mice are described in Appendix A. Due to the low sample size for humans, the study was underpowered to perform SD calculations.

**Figure 4 biomolecules-14-00582-f004:**
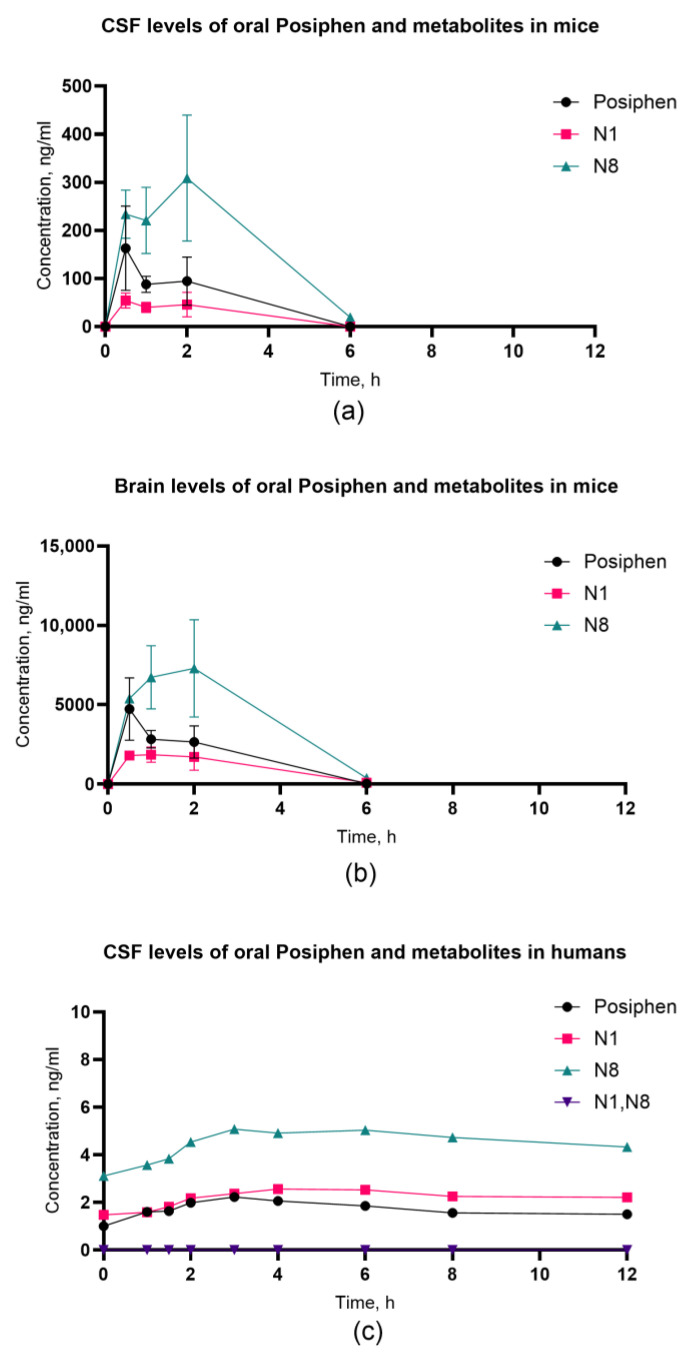
Plasma concentrations of oral Posiphen and metabolites presented as Cmax mean values with standard deviation (SD) in mice (n = 7)—(**a**) CSF; (**b**) brain—and in humans (n = 4)—(**c**) CSF. The SD and %CV values for each time point in mice are described in Appendix A. Due to the low sample size for humans, the study was underpowered to perform SD calculations.

**Figure 5 biomolecules-14-00582-f005:**
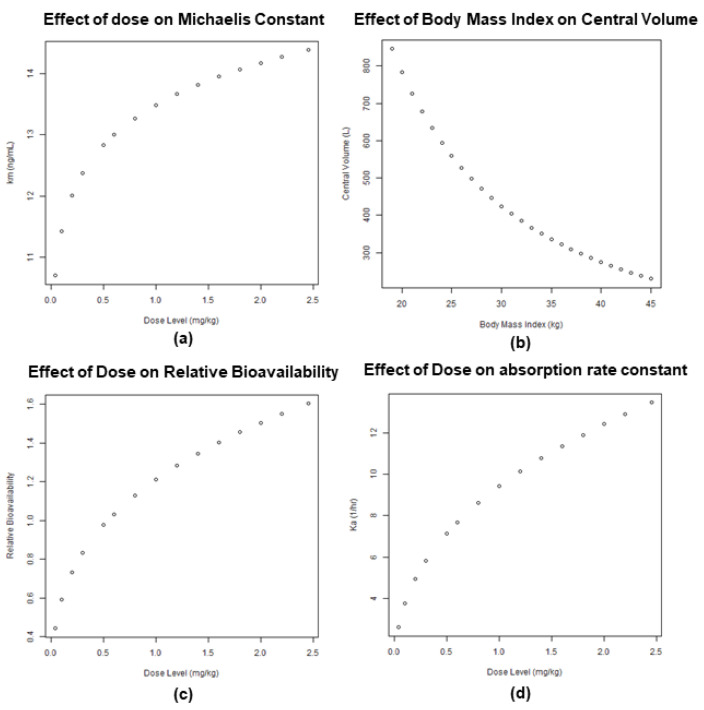
Effect of dose on multiple parameters: (**a**) Michaelis constant; (**c**) relative bioavailability; (**d**) absorption rate constant; (**b**) effect of BMI on central volume.

**Table 1 biomolecules-14-00582-t001:** Summary of Posiphen PK parameters across different species—mouse (*n* = 7), rat (*n* = 6), dog (*n* = 12), and human (*n* = 4). The following parameters are presented: Cmax (ng/mL), Tmax (h), T1/2 (h), AUC (0–8 h) (ng × h/mL).

Posiphen PO	Cmax	Tmax	T1/2	AUC (0–8 h)
Mouse [65 mg/kg]	939 ng/mL	0.5 h	0.562 h	1707 ng × h/mL
Rat [40 mg/kg]	1180 ng/mL	0.5 h	1.03 h	1140 ng × h/mL
Dog [20 mg/kg]	402 ng/mL	1.18 h	1.96 h	756 ng × h/mL
Humans [1.1 mg/kg]	120 ng/mL	1.45 h	4.04 h	310 ng × h/mL

**Table 2 biomolecules-14-00582-t002:** Pharmacokinetic parameters of Posiphen and N1, N8 metabolites in plasma, CSF, and brain in mouse and human samples. Presented parameters include Cmax (ng/mL), Tmax (h), T1/2 (h), and AUC (0-t) (h × ng/mL).

	Mouse				Human			
	Cmax	Tmax	T1/2	AUC (0-t)	Cmax	Tmax	T1/2	AUC (0-t)
	Plasma				Plasma			
Posiphen	939	0.562	0.5	1707	117	3.73	1.5	564.71
N1	554	0.717	1.25	1403	24.9	5.80	2	213.26
N8	1931	0.863	1.5	5334	30.4	7.03	2	259.99
N1, N8	ND	ND	ND	ND	3.66	13.61	3	36.89
	CSF				CSF			
Posiphen	163	2.32	0.5	193	2.23	17.26	3	20.70
N1	54.4	9.09	0.5	80	2.56	25.31	4	26.73
N8	309	ND	2.0	856	5.08	27.90	3	54.92
	Brain							
Posiphen	4737	0.74	0.5	8202				
N1	1854	1.10	1.0	5402				
N8	7293	ND	2.0	20,766				

## Data Availability

All reports containing data are available in the reference section of the paper or upon request from Annovis Bio.

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
