# Peer review of "Comparative Analysis of Posiphen Pharmacokinetics across Different Species—Similar Absorption and Metabolism in Mouse, Rat, Dog and Human"

_biomolecules, 2024, doi:10.3390/biom14050582_

Round 1
Reviewer 1 Report
Comments and Suggestions for Authors
This manuscript considers species-specific considerations useful in understanding development and translational aspects of posiphen and metabolite pharmacokinetics. The studies add to the extensive evaluations done in animal models and human studies of posiphen.
Improvements to the manuscript could be made by the addition of supplemental materials. More details of the analytical methods, doses, sample size, bioavailability, and statistical evaluation of the animal studies including error bars on graphs would be useful.
A more thorough description of the results from previous animal PK studies and a comparison with the current results would be useful. Also useful would be a proposed explanation of the observation of the persistence of drug and its metabolites in brain and CSF.
Author Response
Dear Reviewer 1,
We sincerely appreciate the time you took to provide a detailed and constructive feedback on our manuscript. We have carefully considered each of your comments and have made corresponding revisions, which are outlined below. Additionally, in the resubmitted version of the manuscript, we highlighted changes in grey. We believe these improvements have enhanced the manuscript, and we are hopeful that it is now prepared for the subsequent stages of publication. Should you have any further inquiries or recommendations, we welcome your input.
- Improvements to the manuscript could be made by the addition of supplemental materials.
We have added supplementary materials that include: 1) a detailed methodology for the popPK portion of the study; 2) tables with statistical values for presented data; 3) additional graphs for N1/N8 metabolites in rats and dogs, that are not shown in the main body of the text, but are referenced to in the Discussion.
- More details of the analytical methods, doses, sample size, bioavailability, and statistical evaluation of the animal studies including error bars on graphs would be useful.
We expanded our Methods by including all requested points. Additionally, we added error bars to our graphs for all animal species. However, for humans, the n-number was low and, hence, the study was underpowered for to provide accurate error calculations. We disclosed this information in the Methods and Figure legends where appropriate.
- A more thorough description of the results from previous animal PK studies and a comparison with the current results would be useful. Also useful would be a proposed explanation of the observation of the persistence of drug and its metabolites in brain and CSF.
Our former published studies were largely focused on the drug’s safety and efficacy profiles, while the PK studies were never published and limited to our internal reports. This is the first study to draw results from these reports. However, we did add additional details in Discussion on the comparison of the observed PK differences and provided more information on the drug’s persistence in the brain/CSF and its effect on possible outcomes.
Reviewer 2 Report
Comments and Suggestions for Authors
The manuscript titled "Pharmacokinetics of Posiphen in Mouse, Rat, Dog and Human" presents a comprehensive study on the pharmacokinetic profile of Posiphen, a small molecule with potential applications in treating neurodegenerative diseases. The study's objective is to fill a gap in the literature by examining the metabolic profiles and the breakdown of Posiphen into its primary metabolites, N1 and N8, across different species. The findings reveal species-specific behavior of Posiphen and its metabolites, with implications for clinical translation and human dosing.
Title: The title should be revised to more accurately reflect the comprehensive content of the manuscript, especially emphasizing the comparative pharmacokinetic analysis across different species and the implications for clinical translation.
Abstract: The abstract should include a brief description of the main methods used in the study, such as the pharmacokinetic study design, the use of Posiphen and its metabolites, and the population pharmacokinetic (Pop PK) analysis. The conclusion part should clearly state the major findings and contributions of the work, highlighting the significance of the results.
Keywords: N1 and N8, being specific metabolites of Posiphen, may not be universally recognized by all readers. Consider using more general keywords that reflect the study's focus, such as "pharmacokinetics," "metabolism," "neurodegenerative diseases," and "species comparison."
Materials and Methods: The manuscript should provide a detailed description of the experimental procedures, including the identification of Posiphen metabolites, the conduct of CYP phenotyping experiments, and the methodology for the Pop PK simulation. This will help readers understand the basis for the results presented.
Statistical Analysis: The manuscript should include a section on statistical analysis, detailing the methods used to analyze the data, the statistical tests applied, and the level of significance used to interpret the results.
Results: There should be a clear connection between the Pop PK analysis and other results presented in the manuscript. The relationship between these findings and the overall conclusions should be explicitly discussed.
Table 1: The title of Table 1 should be revised for clarity, and it should accurately represent the data presented.
Data Presentation: Figures and tables should be revised to include essential information such as the number of samples, variability, and other relevant details that provide context for the data. This will improve the quality and interpretability of the results.
Discussion: The discussion should include an in-depth analysis of the pharmacokinetic characteristics of Posiphen across different species, with a focus on how these differences might influence dosing and safety in clinical settings. A more explicit comparison of the PK profiles between species and the implications for drug development and clinical application are necessary.
Overall, the manuscript should be reorganized to a thorough revision to address these points, ensuring that the methods are clearly described, the results are well-presented and supported by statistical analysis, and the discussion provides a comprehensive interpretation of the findings in the context of the broader field of pharmacokinetics and drug development.
Author Response
Dear Reviewer 2,
Thank you for your insightful feedback on our manuscript. We sincerely appreciate the time you invested in providing such a thorough analysis. Your constructive comments have been invaluable in enhancing the quality of our work. We have now addressed each of your points and highlighted the changes in the resubmitted manuscript file. Should any further questions or suggestions arise, please do not hesitate to reach out.
Title: The title should be revised to more accurately reflect the comprehensive content of the manuscript, especially emphasizing the comparative pharmacokinetic analysis across different species and the implications for clinical translation.
We updated the title to better reflect the summary of the manuscript.
Abstract: The abstract should include a brief description of the main methods used in the study, such as the pharmacokinetic study design, the use of Posiphen and its metabolites, and the population pharmacokinetic (Pop PK) analysis. The conclusion part should clearly state the major findings and contributions of the work, highlighting the significance of the results.
Methodology and popPK details were added in the Abstract.
Keywords: N1 and N8, being specific metabolites of Posiphen, may not be universally recognized by all readers. Consider using more general keywords that reflect the study's focus, such as "pharmacokinetics," "metabolism," "neurodegenerative diseases," and "species comparison."
This is a great suggestion, the keywords are now updated.
Materials and Methods: The manuscript should provide a detailed description of the experimental procedures, including the identification of Posiphen metabolites, the conduct of CYP phenotyping experiments, and the methodology for the Pop PK simulation. This will help readers understand the basis for the results presented.
A comprehensive update to the Methods part has been made by including details on analytical procedures, animal/human groups, CYP phenotyping, popPK, and statistics. Additionally, we included a supplementary material that thoroughly describes the popPK simulation.
Statistical Analysis: The manuscript should include a section on statistical analysis, detailing the methods used to analyze the data, the statistical tests applied, and the level of significance used to interpret the results.
Thank you for this valuable insight. We included the information on statistical analysis and added error bars in the graphs, where appropriate. Additionally, we provided supplementary materials with statistical values for each presented graph.
Results: There should be a clear connection between the Pop PK analysis and other results presented in the manuscript. The relationship between these findings and the overall conclusions should be explicitly discussed.
We improved the flow of the text to better reflect the connection between our animal studies and the transition into popPK studies. The underlying goal was to analyze the level of variability of Posiphen's metabolism not only between animal species, but also to check if it is reflected across clinical subtypes. This information is now provided in the Results and Discussion.
Table 1: The title of Table 1 should be revised for clarity, and it should accurately represent the data presented.
The Table 1 title has been updated.
Data Presentation: Figures and tables should be revised to include essential information such as the number of samples, variability, and other relevant details that provide context for the data. This will improve the quality and interpretability of the results.
The Figures have been updated with more relevant information including error bars, n-number for each group, and statistics.
Discussion: The discussion should include an in-depth analysis of the pharmacokinetic characteristics of Posiphen across different species, with a focus on how these differences might influence dosing and safety in clinical settings. A more explicit comparison of the PK profiles between species and the implications for drug development and clinical application are necessary.
The Discussion part has been rewritten to include a more comprehensive overview of the study and its possible impact on clinical implications as well as the relevance of preclinical models in translation settings for Posiphen.
Overall, the manuscript should be reorganized to a thorough revision to address these points, ensuring that the methods are clearly described, the results are well-presented and supported by statistical analysis, and the discussion provides a comprehensive interpretation of the findings in the context of the broader field of pharmacokinetics and drug development.
We appreciate this thorough feedback on how to improve the quality of our manuscript. Overall, changes included a more comprehensive approach to the way we present data and discuss its significance. We hope that made alterations shape our paper for the next stages of publication.
Round 2
Reviewer 1 Report
Comments and Suggestions for Authors
The revised manuscript has beeb improved
Author Response
Dear Reviewer,
We appreciate the time you took to review our revised manuscript. Your feedback and acknowledgement of improved details in the resubmitted version bring us one step closer to its publication.
Don't hesitate to let us know in case you have any additional comments.
Reviewer 2 Report
Comments and Suggestions for Authors
The author has endeavored efforts to respond to the reviewer's questions and a serious revision has been made. A great improvement in quality was accomplished. The main issues currently should be addressed:
1. The exploration of interspecies differences and similarities in pharmacokinetics is highly meaningful for the research of new drugs, as it can bridge the gap from the bench to the bedside. However, the authors only conducted pharmacokinetic studies of a single oral administration of posiphen in mice, rats, beagle dogs, and humans, and relatively in-depth compared the pharmacokinetic characteristics between mice and humans. It can be noted that there are significant differences in the metabolic profiles and exposure characteristics of the parent drug between mice and humans. The author has neither compared the metabolite spectrum of posiphen nor the in vitro metabolic stability across different species. How can the conclusion that posiphen has similar absorption and metabolism across different species be drawn? The data and core of the entire article are not in a rigorous logically manner.
2. There are still plenty of errors in the writing of the article, including issues with capitalization, subscripts, etc. The quality of the figures needs to be improved, including ensuring that the horizontal axis of the PK curves remains consistent, and so on.
Comments on the Quality of English LanguageThe manuscript would benefit from further refinement to enhance clarity and precision in the presentation of ideas.
Author Response
Dear Reviewer,
We sincerely appreciate your thorough feedback and suggestions for enhancing our manuscript. Below, we have addressed each of the points you raised:
- The exploration of interspecies differences and similarities in pharmacokinetics is highly meaningful for the research of new drugs, as it can bridge the gap from the bench to the bedside. However, the authors only conducted pharmacokinetic studies of a single oral administration of posiphen in mice, rats, beagle dogs, and humans, and relatively in-depth compared the pharmacokinetic characteristics between mice and humans.
Our response: We agree with the reviewer that we are only looking at one dose in rats and dogs. However, to effectively compare pharmacokinetics (PK) across different species, it is essential to minimize variables, given the inherent variability between the animals and humans. Administering a single dose, as opposed to multiple doses, eliminates the confounding factors associated with timing and repeated exposure. This allows to improve the accuracy and reliability of the comparative analysis.
- It can be noted that there are significant differences in the metabolic profiles and exposure characteristics of the parent drug between mice and humans. The author has neither compared the metabolite spectrum of posiphen nor the in vitro metabolic stability across different species.
Our response: We agree with the reviewer that in plasma we see significant differences between mice and humans. We would like to point out, that while the liver enzymes in mice cause Posiphen to be metabolized faster and to a greater extent than in humans, the CSF levels of active Posiphen and metabolites are comparable in both species. Additionally, we would like to emphasize that this study exclusively examines the in vivo effects of Posiphen, building upon previous in vitro research documented in other publications. With our ongoing clinical trials yielding real human data, the objective was to contrast these clinical findings with our previous animal studies. In the Results and Discussion sections of the manuscript, we provided a comparison of Posiphen's metabolite spectrum. However, a more comprehensive analysis has been added in the revised Discussion.
- How can the conclusion that posiphen has similar absorption and metabolism across different species be drawn? The data and core of the entire article are not in a rigorous logically manner.
Our response: We acknowledge that our earlier conclusions may have been overly optimistic. While we did not explicitly assert equivalence in absorption and metabolism of Posiphen between species, we did suggest some similarity in their pharmacokinetic (PK) profiles. Our reasoning was grounded in rapid peak concentration (under 2 hours) and fast clearance (under 12 hours) in plasma, contrasted to slower peak concentration and longer clearance in brain/CSF. We also wanted to point out that Posiphen’s metabolites have TINAP activity, and the key metabolite in brain/CSF is N8, which only shows TINAP and absolutely no AChEI activity. Additionally, we considered the lower levels of N1 as another point of similarity, due to a potentially reduced toxicity. However, we agree that this analysis may have oversimplified the complexities involved, leading to inaccuracies in our conclusion. While these observations may imply the translational potential of a mouse model, they do not necessarily validate this model as the best one for preclinical examination of Posiphen. We have revised the Conclusion section to better align with the evidence provided by our data.
- There are still plenty of errors in the writing of the article, including issues with capitalization, subscripts, etc. The quality of the figures needs to be improved, including ensuring that the horizontal axis of the PK curves remains consistent, and so on.
Our response: We revised the grammar and punctuation in the article and updated the figures accordingly.